# BM-MSCs mitigate lung injury in a rat model of decompression sickness

Chen Lu[1☉], Daqian Gu[2☉], Hao Chen[1], Liang Chen[1], Jie Chen[1], Yuwei Weng[1], Xianliang Lin[1]*

1 Department of Cardiology, Fuzong Clinical Medical College of Fujian Medical University, Fuzhou, 350001, China, 2 Department of Critical Care Medicine, Jinling Hospital, Affiliated Hospital of Medical School, Nanjing University, Nanjing, 210002, China

☉ These authors contributed equally to this work.
* lxliang1531@163.com

## Abstract

Decompression sickness is a fatal disease worldwide. Therefore, to find a prophylactic modality for decompression sickness is urgently required. Bone marrow derived mesenchymal stem cells exhibit effectiveness in antioxidant, anti-inflammation, and decrease cell death; while its effects on decompression sickness remains unclear. This study aimed to further investigate the mechanisms of decompression sickness induced lung injury, as well as effects of bone marrow derived mesenchymal stem cells on decompression sickness induced lung injury and explore the role of oxidative stress, inflammation and cell death play in this disease. The study involved Sprague-Dawley rats age at 8−10 weeks weighting 350±10g. Acute lung injury was induced by decompression hyperbaric chamber. A dose of bone marrow derived mesenchymal stem cells ($2 \times 10^6$ cells) was given to rats one day prior to the start of decompression. Lung injury severity was estimated by determining lung damage scores, pulmonary oxidative, inflammatory factors and cell death. In bone marrow derived mesenchymal stem cells treated rats, the morbidity and mortality of decompression markedly decreased. The increases of protein IL-1 and IL-6 in BALF and lung wet/dry ratio and lung injury score were alleviated. The ROS, CAT, SOD, and MDA activities and GSH levels were significant attenuated ($P < 0.05$). The pyroptosis and nerroptosis were significant mitigate ($P < 0.05$). Based on the results, bone marrow derived mesenchymal stem cells is an potential efficient and safe prophylactic modality protect rats from decompression induced acute lung injury.

## Introduction

Decompression sickness (DCS) is a systemic and mortal disease [1]. It has been the most serious danger for Self-Contained Underwater Breathing Apparatus (SCUB) divers [2]. In DCS, gas exchange disorder and respiratory distress always represent

**Data availability statement:** All relevant data are within the paper and its Supporting information files.

**Funding:** The present work was supported by the grants CLB19J038 from Military Logistics Research Program China and recipient of the grants is Xianliang Lin.

**Competing interests:** The authors have declared that no competing interests exist.

as common symptoms [3]. In pathological manifestations of lung in DCS, the alveolar epithelial cells show hydropic degeneration and disintegration whilst the pulmonary capillary endothelium is often ruptured, which might explain for the lung edema and hemorrhage [4]. Although the exact mechanisms that lead to lung injury are still not clear, there are some possible effective strategies to prevent or limit its progression [5,6]. Thus, a better understanding of the mechanisms involved in the progression of DCS-induced lung injury and finding appropriate treatment are essential.

DCS can trigger the acute stimulation of immune system, and subsequently induce inflammation [7]. The initiation of inflammation might be attributed to increased oxidative stress, including overproduction of reactive oxygen species (ROS), since a rapid increase in ROS can activate nuclear factor κB (NF-κB) [8]. A former study has shown that swollen mitochondria exists in granular pneumocytes under DCS, which is the characteristic of enhanced mitochondrial ROS formation [4]. Similarly, both in vivo and in vitro studies demonstrated that DCS increased content of ROS [9,10]. Uncontrolled ROS may trigger mitochondrial permeability transition pore (mPTP) induction within individual mitochondria in intact cell systems [11]. The phenomenon of ROS-triggering of the mPTP associated with further stimulation of ROS formation has been termed "ROS-induced ROS release" (RIRR) [12].ROS overproduction between mitochondria cause a positive-feedback mechanism resulting in an elevated production of ROS that could be propagated throughout the cell and subsequently destroy the mitochondrial dynamics and other pathways, and eventually induce cell death [13]. Besides, persistent cell death might further exacerbate inflammation and oxidative stress. Some types of cell death such as necrosis and apoptosis have been shown in DCS injured brain tissue [14]. However, what types of cell death are involved in DCS-induced lung injury is still unknown.

Bone marrow-derived mesenchymal stem cells (BM-MSCs) have been proved as a potential cell-based therapy for lung disease such as chronic obstructive pulmonary disease (COPD), lipopolysaccharide (LPS)-induced lung injury and so on [15,16]. Because BM-MSCs are thought to be immune-privileged and protected from rejection [17], it has been permitted to be used in allo-transplantation in some clinical researches [18,19]. A study has demonstrated that BM-MSCs could engraft into the injured lung [20]. In addition, BM-MSCs display anti-inflammation and anti-oxidative stress effects in lung injuries and other diseases [21,22]. For example, in Tarek Khamis et al study, BM-BMSCs have been demonstrated decrease both the mean fold change of the relative mRNA expression of the renal proinflammatory markers such as NFKβ, IL1β, TNFα, and IL6 and the mean fold change of the relative mRNA expression of renal proapoptotic markers Fas, FasL, P53, caspase-3, BAX, and BAX/BCL2, meanwhile significantly upregulating the anti-apoptotic marker BCL2 compared with the diabetic group [23]. Asmaa Adel et al. showed that the treatment of CCl4-injected rats with rats and mice BM-MSCs significantly elevated SOD and GST levels. However, whether BM-MSCs could reduce inflammation, oxidative stress and cell death in DCS-induced lung injury is seldom investigated.

In this study, we pretreat DCS model with BM-MSCs in different time points to evaluate the effects of prevention and treatment. We analyzed the morbidity, mortality, lung injury, inflammation and oxidative stress. In addition, to study the types of pulmonary cell death in DCS, we detected the expression of markers for apoptosis, pyroptosis, necroptosis and ferroptosis. The present study aimed to elucidate the effect of BM-MSCs on DCS-induced lung injury.

## Methods

### Animals

A total of 252 healthy Sprague-Dawley (SD) rats age at 8–10 weeks weighting $350 \pm 10$ g were purchased from Shanghai Slac Laboratory Animal Co. Ltd and bred in an AAALAC-accredited facility. The animals were housed in a controlled environment ($20 \pm 2$ °C, 12h/12h light/dark cycles), with free access to water and standard rodent chow. All experimental procedures were approved by the Animal Care and Use Committee of the 900th Hospital of Joint Logistics Support Force. All experiments conformed to the guidelines for the ethical use of animals, and all efforts were made to minimize animal suffering and reduce the number of animals used.

### DCS model

For DCS model, the rats were divided into three groups, and they were placed in a hyperbaric chamber (Hongyuan Oxygen Industry Co., Ltd, Yantai, China). For DCS1 group, the chamber was pressurized to 6 bar in 5 min at a speed of 1 bar/min and maintained for 90 min [5], after that the chamber was decompressed to 1 bar at a speed of 2 bar/min. For DCS2 group, the chamber was pressurized to 6 bar in 5 min at a speed of 1 bar/min and maintained for 90 min, after that the chamber was decompressed to 1 bar at a speed of 1 bar/min. For DCS3 group, the chamber was pressurized to 7 bar in 5 min at a speed of 1.2 bar/min and maintained for 90 min, after that, the chamber was decompressed to 1 bar at a speed of 2 bar/min. The rats were divided into five groups, and respectively named as Vehicle, DCS, DCS + BM-MSCs 7d, DCS + BM-MSCs 3d and DCS + BM-MSCs 1h. The rats in Vehicle group were treated with saline vehicle (control, 200 μL, tail vein injection); the rats in DCS group were treated with saline vehicle (200 μL, tail vein injection) and then experienced DCS modeling; the rats in MSCs-7d group were treated with BM-MSCs ($2 \times 10^6$ cells, 200 μL saline as vehicle, tail vein injection) 7 days before DCS modeling [24,25]; the rats in MSCs-3d group were treated with BM-MSCs ($2 \times 10^6$ cells, 200μ L, saline as vehicle, tail vein injection) 3 days before DCS modeling; the rats in DCS + BM-MSCs 1h group were treated with BM-MSCs ($2 \times 10^6$ cells, 200 μL, saline as vehicle, tail vein injection) 1 hour (h) before DCS modeling. During the exposure, the chamber was ventilated continuously to avoid carbon dioxide ($CO_2$) retention and the temperature was controlled at $25 \pm 2$°C. Following the decompression procedure, rats were observed for DCS related behaviors within 2 h by a member of staff who was blinded to the treatments and possess certificate of Laboratory Animal Practitioner Training. Any of the following symptoms was regarded as presence of DCS: respiratory distress, walking difficulties, fore and/or hind limb paralysis, rolling, convulsions or death [3]. After the 2 h observation period, all survived rats were intraperitoneally anesthetized with 3% pentobarbital sodium (1.5 mL·kg$^{-1}$) and then blood, bronchoalveolar lavage fluid (BALF) and lung tissues were sampled for further analysis. The wet/dry weight ratio of the right upper lobes of lung was also calculated.

### Isolation and culture of BM-MSCs

BM-MSCs were isolated and expanded according to a previously described procedure [26]. In brief, bone marrow was flushed from the femoral and tibia with phosphate-buffered saline (PBS). Then, the bone marrow was passed through a 70 mm strainer and centrifugated at 1,200 rpm for 5 min. Thereafter, the cell palet were extracted and resuspended in Dulbecco's modified Eagle's medium (DMEM) (Gibco, Shanghai, China) supplemented with 20% fetal bovine serum (FBS) and 1% penicillin/streptomycin medium. Third-passage MSCs were used for different treatments.

## Phenotypes and differentiation of BM-MSCs

To analyze the cell surface markers for BM-MSCs, cells were detached with trypsin and ethylenediamine tetraacetic acid, washed and re-suspended in PBS with 5 mM EDTA. Then cells were stained with CD105, CD90, CD73, CD34, CD45 and HLA-DR for 1 hour at 4°C. After that, the cells were washed twice and stained with APC, FITC and PE conjucted antibodies. At last, the cells were washed and analyzed by flow cytometry [27].

To induce osteogenic differentiation, cells were treated with the osteogenic medium (complete medium with 0.1 mmol·L$^{-1}$ dexamethasone, 10 mmol·L$^{-1}$ b-glycerol phosphate and 50 mg·mL$^{-1}$ ascorbate-2-phosphate) for 18 days. The medium was changed every 3 days. After induction, cells were fixed with 4% paraformaldehydes or 20 min and washed with PBS (5 min, 3 times), and then stained with 1% alizarin red solution (Solarbio, Beijing, China) for 10 min at 37°C to examine the mineral nodule deposition [28].

To induce chondrogenic differentiation, cells were cultured in complete medium supplemented with 100 mg·mL$^{-1}$ sodium pyruvate, 10 ng·mL$^{-1}$ transforming growth factor-β1, 100 nmol·L$^{-1}$ dexamethasone, 1% insulin-transferrin-selenium and 100 mg·mL$^{-1}$ ascorbate-2-phosphate for 14 days. After induction, cells were fixed with 4% paraformaldehydes for 20 min and washed with PBS (5 min, 3 times), and then stained with Alcian Blue (Solarbio) for 30 min to evaluate deposition of glycosaminoglycans [28].

To induce adipogenic differentiation, cells were cultured in complete medium with 0.5 mmol·L$^{-1}$ isobutylmethylxanthine, 0.25 mmol·L$^{-1}$ dexamethasone, 10 mmol·L$^{-1}$ insulin and 50 mmol·L$^{-1}$ indomethacin for 10 days. After that, cells were fixed with 4% paraformaldehydes for 20 min and washed with PBS (5 min, 3 times), and then stained with oil red O solution (Solarbio) for 45 min to examine lipid droplets in cytoplasm. All the staining results were observed under an inverted microscope (Olympus, Tokyo, Japan) and photographed [28].

## Lung histopathology

The right lower lobes of lung samples were fixed and then dehydrated in increasing concentrations of ethanol. After that, they were cleared in xylene and embedded in paraffin. The samples were cut into 5 μm thick sections followed by staining with haematoxylin and eosin. Morphological damages were scored to evaluate the degree of lung damage. These histopathologic damages included alveolar congestion, hemorrhage, neutrophil infiltration into the airspace or vessel wall, and thickness of alveolar wall/hyaline membrane formation [29] Pathological scoring based on the injury area of involvement was according to a 5 point scale as follows: 0 = minimal damage, 1 = mild damage, 2 = moderate damage, 3 = severe damage, and 4 = maximal damage [30]. Quantification was conducted by an investigator who was blinded to the group information of the samples.

## Lung ROS detection

The rat lung tissues were quickly frozen, cut to a thickness of 8 μm at an optimized cutting temperature, and mounted on glass slides. ROS [31] was stained by DHE probe as red and cell nucleus was stained by DAPI as blue. The concentration of DHE in the lung was calculated by the fluorescence intensity using Image J, according to the manufacturer's protocol (Beyotime Institute of Biotechnology, China).

## BALF analysis

Total protein level in BALF was determined according to the Bradford protein assay kit (Solarbio) by measuring the absorption at 595 nm with a spectrum photometer. The BALF was centrifuged at 3,000 rpm for 20 min at 4°C. The extracted cell pellet was resuspended by 1 mL PBS and used to determine the total cell count through a cell counter. A cell smear was made and stained by Wright-Giemsa staining to confirm the neutrophil percentage.

## Assay of oxidative stress and inflammatory markers

Lung tissues were homogenized in assay buffer. Subsequently, the tissue homogenate was used to detect the content of glutathione (GSH) and myeloperoxidase (MPO), malondialdehyde (MDA), superoxide dismutase (SOD), catalase (CAT) activities by commercial reagent kits (Jiancheng Bioengineering Institute, Nanjing, China) according to instructions, respectively. Values were normalized by tissue protein concentration. The fresh blood samples were placed under room temperature for 1 h, and then the samples were centrifuged under 4 °C, 2,000g for 10 min. The supernatant was extracted as serum for detection. The serum levels of Tumor Necrosis Factor-α (TNF-α), Interleukin-1β (IL-1β) and Interleukin-6 (IL-6) were measured using the Elisa Assay Kits (Beyotime Institute of Biotechnology, Nantong, China).

## Immunoblotting

The lung tissues were lysed in RIPA buffer, supplemented with protease inhibitors, and denatured with loading buffer. The nuclear and cytoplasmic fractions were also denatured with loading buffer. The protein samples were collected and stored at – 20° C until use. The protein samples were separated by SDS-PAGE with 10−15% polyacrylamide gel and then electroblotted onto nitrocellulose membranes (Amersham Life Science, Arlington, TX). The blots were blocked in Tris-buffered saline containing 5% nonfat dry milk for 1 h at room temperature with constant shaking and then incubated with primary antibodies (Cleaved- Caspase 3, Bcl-2, Caspase-1, NLRP3, RIPK3, MLKL, GPX4, ACSL4, β-actin) overnight at 4°C. The secondary antibodies (Goat anti-Rabbit IR Dye 800, Donkey anti-Goat IR Dye 800, Goat anti-Mouse IR Dye 800) were used to bind their respective primary antibody at room temperature for 1 hour. The bound complexes were detected using the Odyssey Infrared Imaging System (Li-Cor Biosciences). The images were analyzed through the Odyssey Application Software to obtain the integrated intensities.

## Statistical analysis

All data are expressed as mean ± SD, unless otherwise stated. GraphPad-Prism 9.0 and SPSS 17.0 were used to perform the statistical analyses. Student's unpaired t-test was used to compare 2 independent groups. Continuous variables were tested for normal distribution with the Kolmogorov-Smirnov test. Incidence of DCS was compared by Chi-square test. Survival rates were compared using the log-rank test. Values of $p < 0.05$ were considered significant.

## Results

### Identification of BM-MSCs

Isolated rat BM-MSCs were characterized by flow cytometry and their differentiation capacity. Flow cytometry analysis showed that the cells were positive for CD105, CD73 and CD90, whereas negative for CD45, CD34 and HLA-DR. The results conformed to the characteristics of MSCs (Fig 1). To provide a clear visual reference for background signal discriminationwe have incorporated isotype control overlays in Fig 2. Phase-contrast microscopy images of BM-MSCs at passage 3, demonstrating their spindle-shaped morphology (Fig 3A). After osteogenic induction, alizarin red staining showed that calcium-rich extracellular matrix in the cells (Fig 3B); Alcian blue staining (Fig 3C) confirmed sulfated glycosaminoglycan (sGAG) deposition in the extracellular matrix, indicating successful chondrogenic differentiation of BM-MSCs. After adipogenic induction, oil Red O staining showed that the red lipid droplets were distributed in and between the cells (Fig 3D). These results demonstrated that isolated cells were BM-MSCs.

### Pretreatment of BM-MSCs improves the survival rate and incidence of DCS

To explore the suitable condition for the DCS rat model, we distributed rats into three groups (Fig 4A) and investigated the effects of chamber pressure and decompression rate on them. DCS2 group rats were treated with slower decompression rate than DCS1 group rats (1 bar/min vs. 2 bar/min), while DCS3 group rats experienced higher chamber pressure

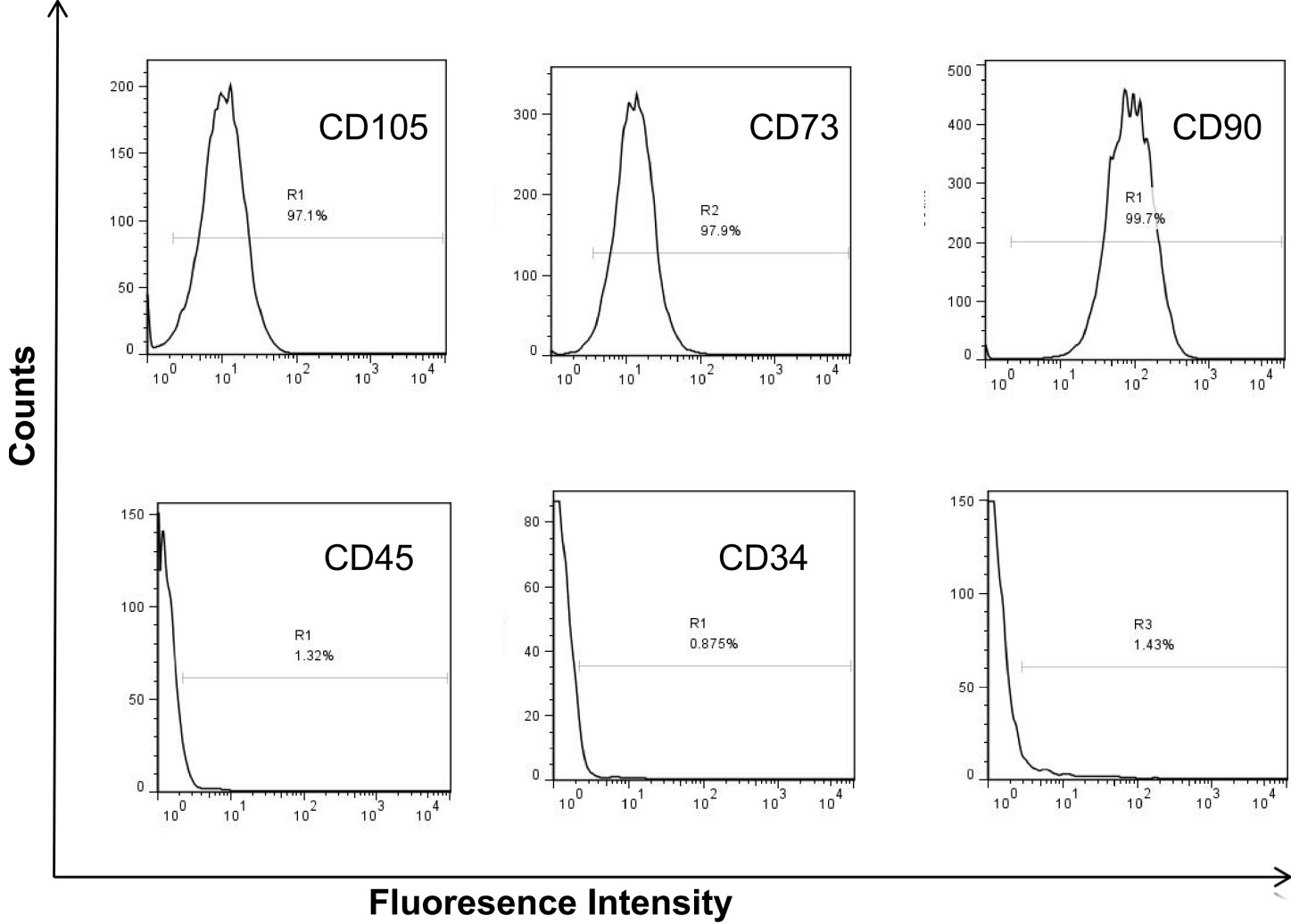

**Fig 1. Characterization of BM-MSC.** Flow cytometry analysis showed that these cells were positive for CD105, CD73 and CD90, but negative for CD34, CD45 and HLA-DR.

than DCS1 group rats (7 bar vs. 6 bar). The results showed that DCS2 group rats experienced less morbidity than DCS1 $0.4583 \pm 0.1309$ (n = 36, 95% CI: 0.1948 to 0.7218, p < 0.05) . DCS1 group decreased more mortality than DCS3 group rats $0.3333 \pm 0.1300$ (n = 36, 95% CI: 0.07159 to 0.5951, p < 0.05) (Fig 4C). Thus, the following DCS model used the experiment condition of DCS1 group rats. To evaluate the influence of BM-MSCs injecting time on DCS model, we treated rats with BM-MSCs 7d, 3d, and 1h before decompression respectively (Fig 4B). We found that injecting BM-MSCs 1h before decompression could significantly reduce morbidity rate $0.4167 \pm 0.1045$ n = 36, 95% CI: 0.2083 to 0.6250, p < 0.05) and mortality rate $0.3333 \pm 0.1027$ (n = 36, 95% CI: 0.1286 to 0.5381, p < 0.05), while injecting 7d and 3d before did not influence it (Fig 4D). Thus, to further investigate the treatment effects of BM-MSCs on DCS rat model, we treated DCS rats with BM-MSCs 1h before decompression in the following experiment. BM-MSCs could decrease the incidence of DCS with time (hazard ratio0.3655, 95% CI 0.1981 to 0.6742) (Fig 4E). Pretreatment with BM-MSCs prolong survival time (hazard ratio 0.2434, 95% CI 0.1051 to 0.5635). (Fig 4F). These results hinted that pretreatment BM-MSCs could protect against DCS.

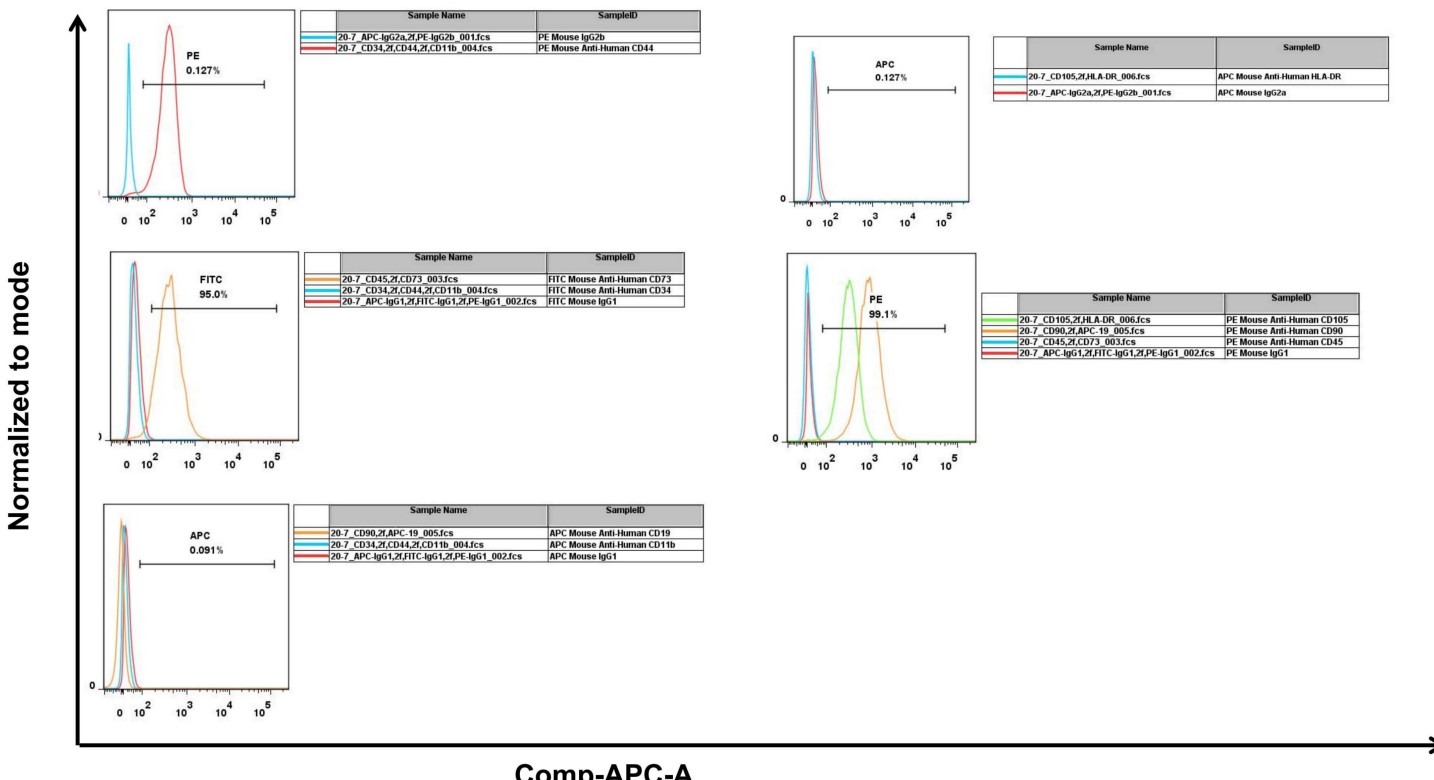

**Fig 2. Isotype control overlays.**

## Pretreatment of BM-MSCs attenuates DCS induced lung injury

Since respiratory dysfunction and lung tissue injury are features of DCS, we wondered whether BM-MSCs could attenuate DCS induced lung injury. The DCS rats in our experiment showed increased W/D ratio and BALF protein, indicating an obvious edema and exudation, in agreement with previous reports. However, pretreatment with BM-MSCs mean decreased W/D ratio −0.8800±0.1975 (n=5, 95% CI: −1.335 to −0.4246, p<0.05) and BALF protein −407.0±64.53 (n=5, 95% CI: 1.335 to −0.4246, p<0.05) compare DCS rats (Fig 5A and 5B). BM-MSCs also ameliorated the DCS-induced lung pathological injury, indicated as reduced alveolar congestion, hemorrhage, neutrophil infiltration, thickness of alveolar wall (Fig 5F). BM-MSCs group reduce mean lung injury score −6.400±0.6000 (n=5, 95% CI: 1.335 to −0.4246, p<0.05) compare with DCS group and statistics graph was shown in (Fig 5C).

## Pretreatment of BM-MSCs attenuates DCS induced lung inflammation

Because inflammation is a common cause for lung injury and rapid decompression could induce the release of inflammatory factors [32], we detected the effect of BM-MSCs on DCS-induced lung inflammation. The results showed that decompression significantly increased the total inflammatory cell and neutrophil counts in the BALF compared with that in vehicle treated rats, but BM-MSCs significantly inhibited the effect t of decompression. BM-MSCs group mean reduce total cells in BALF −4.45*$10^6$±1.10*$10^6$ (n=5, 95% CI: −6.99*$10^6$ to −1.91*$10^{6,}$ p<0.05) and neutrophil counts in BALF −2.28*$10^6$±0.5*$10^6$ (n=5, 95% CI: −3.49*$10^6$ to −1.1*$10^6$, p<0.05) compare with DCS group (Fig 6A and 6B). Meanwhile, we measured the MPO activity to assess the quantification of neutrophil accumulation in tissues. Compared with Vehicle group rats, DCS group rats showed a significant increase in lung MPO activity, but BM-MSCs pretreatment

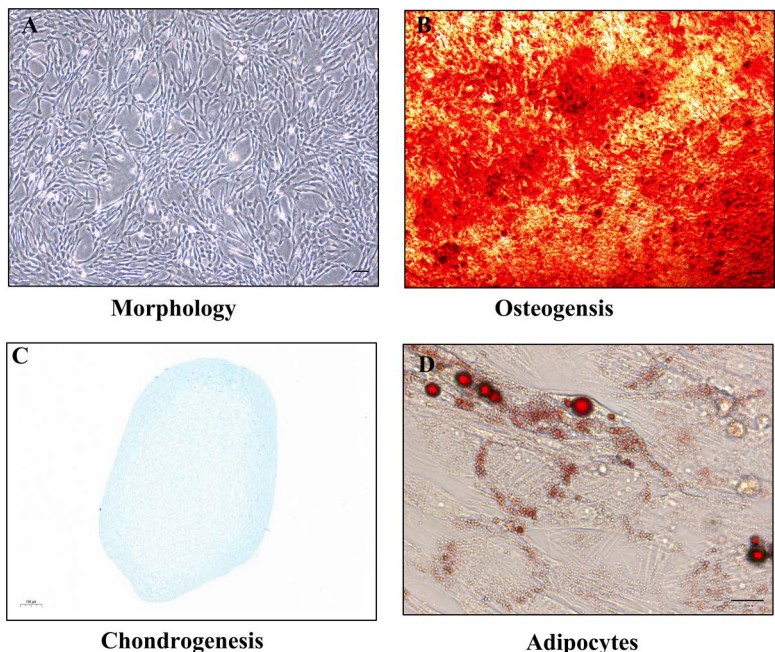

**Fig 3. Identification of BM-MSCs. (A)** The morphology of BM-MSCs was observed under a light microscope (Scale bar = 100 µm). **(B-D)** BM-MSCs were stained with alizarin red (Scale bar = 100 µm), Alcian blue (Scale bar = 100 µm). and oil red O (Scale bar = 20 µm), indicating that could differentiate into osteoblasts, chondrogenic and adipocytes respectively.

reversed it. BM-MSCs mean reduce lung MPO activity $-2.720 \pm 0.5359$ (n = 5, 95% CI: $-3.956$ to $-1.484$, p < 0.05) (Fig 6C). In addition, we detected the expression of inflammatory factors in different groups. We found that BM-MSCs ameliorated the increased serum levels of inflammation-related markers in DCS group rats. BM-MSCs group reduce serum TNF-α-369.4 ± 56.67(n = 5, 95% CI: $-500.1$ to $-238.7$, p < 0.05), serum IL-6–484.6 ± 76.36(n = 5, 95% CI: $-660.7$ to $-308.5$, p < 0.05) and serum IL-1β-209.4 ± 32.76 (n = 5, 95% CI: $-284.9$ to $-133.9$, p < 0.05) (Fig 6D–6F). These results suggested that BM-MSCs might reduce DCS-induced lung injury by decreasing lung and serum inflammation.

## Pretreatment of BM-MSCs inhibits oxidative stress in DCS rats

Excess oxidative stress could activate inflammation response and exacerbate the development of lung injury [33]. Thus, we evaluated the oxidative balance in this experiment. Compared to the Vehicle group rats, the lung ROS production increased in DCS group rats, while BM-MSCs decreased the ROS production induced by decompression-0.6659 ± 0.1114 (n = 5, 95% CI: $-0.9229$ to $-0.4089$ p < 0.05) (Fig 7A–7B). Our results showed that decompression significantly decreased the lung GSH levels together with SOD and CAT activities in vehicle-treated rats, however, BM-MSCs reversed the effects of decompression (Fig 7C–7E). In addition, the lung MDA levels increased to a greater extent in DCS group rats than in Vehicle group rats, but decreased in BM-MSCs-treated DCS rats compared with vehicle-treated DCS rats (Fig 7F). These results indicated that BM-MSCs played an antioxidative role in DCS-induced lung injury.

## The effect of BM-MSCs on different types of lung cell death in DCS rats

Both inflammation and ROS could cause cell death, but what types of cell death are involved in DCS-induced lung injury remains to be further elucidated. In this experiment, we detected the protein expression of the markers for apoptosis (cleaved caspase-3 and Bcl-2, Fig 6A–6B), necroptosis (RIPK3 and MLKL, Fig 8C–8D), pyroptosis (caspase-1 and

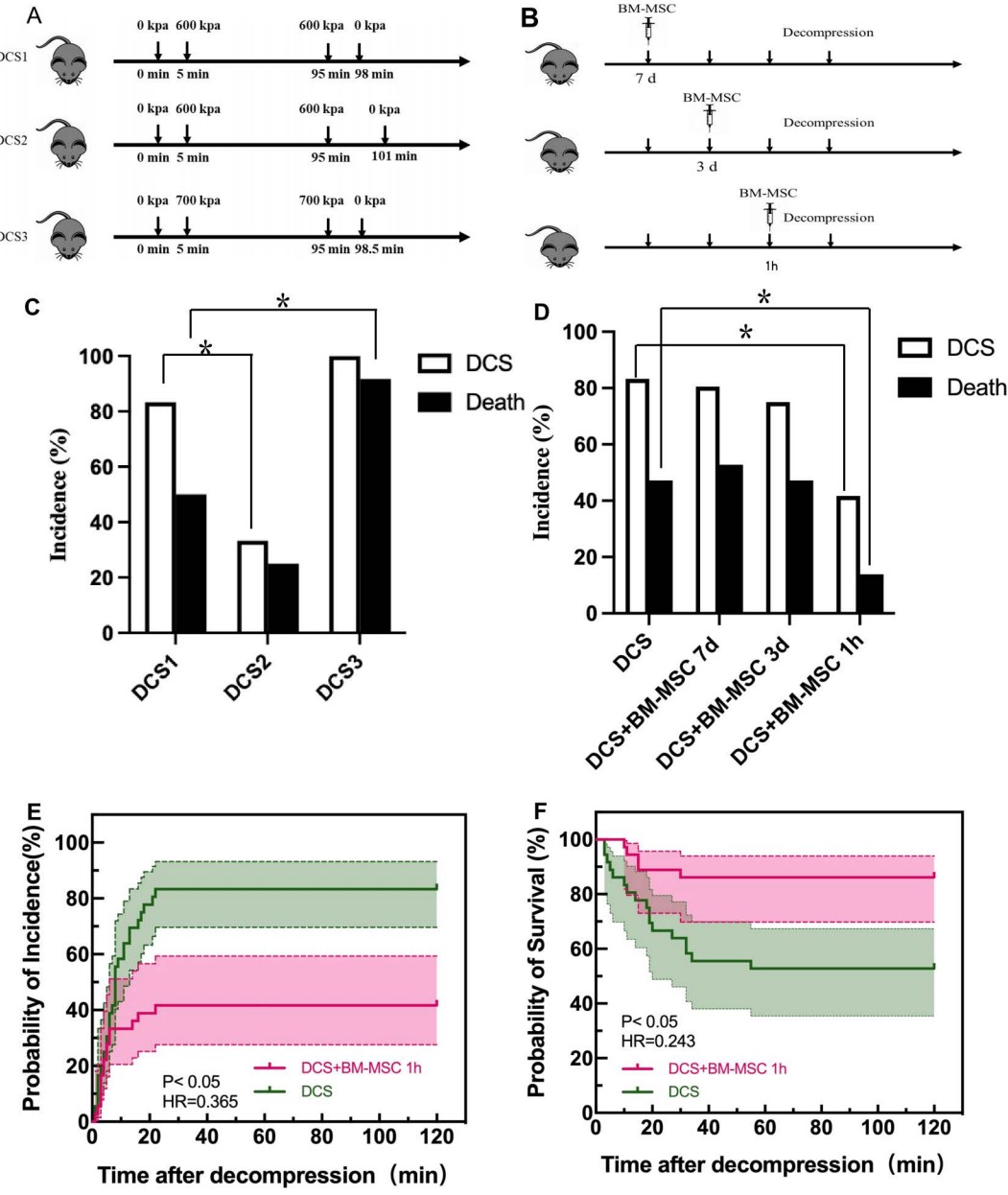

**Fig 4. Pretreatment of BM-MSCs improves the survival rate and incidence of DCS. (A)** Protocol of DCS model. **(B)** Protocol of BM-MSCs injecting. **(C)** The morbidity and mortality of rats under different decompression condition, $n = 36$, * $p < 0.05$ vs. DCS1 group. **(D)** The morbidity and mortality of rats treated with different time, $n = 36$ * $p < 0.05$. **(E)** The incidence of DCS after decompression, $n = 36$ * $p < 0.05$. **(F)** Survival curve of rats after decompression, $n = 36$ * $p < 0.05$.

NLRP3, Fig 8E–8F) and ferroptosis (GPX4 and ACSL4, Fig 8G–8H) in lung tissues in different groups. The results showed that decompression increased the expression of cleaved caspase-3, RIPK3, caspase-1, NLRP3, GPX4, and ACSL4, while decreased the expression of Bcl-2, indicating that decompression might induced apoptosis, pyroptosis and ferroptosis. BM-MSCs pretreatment significantly reduced apoptosis (Bcl-2↑), pyroptosis (Caspase-1↓), and ferroptosis (GPX4↑).

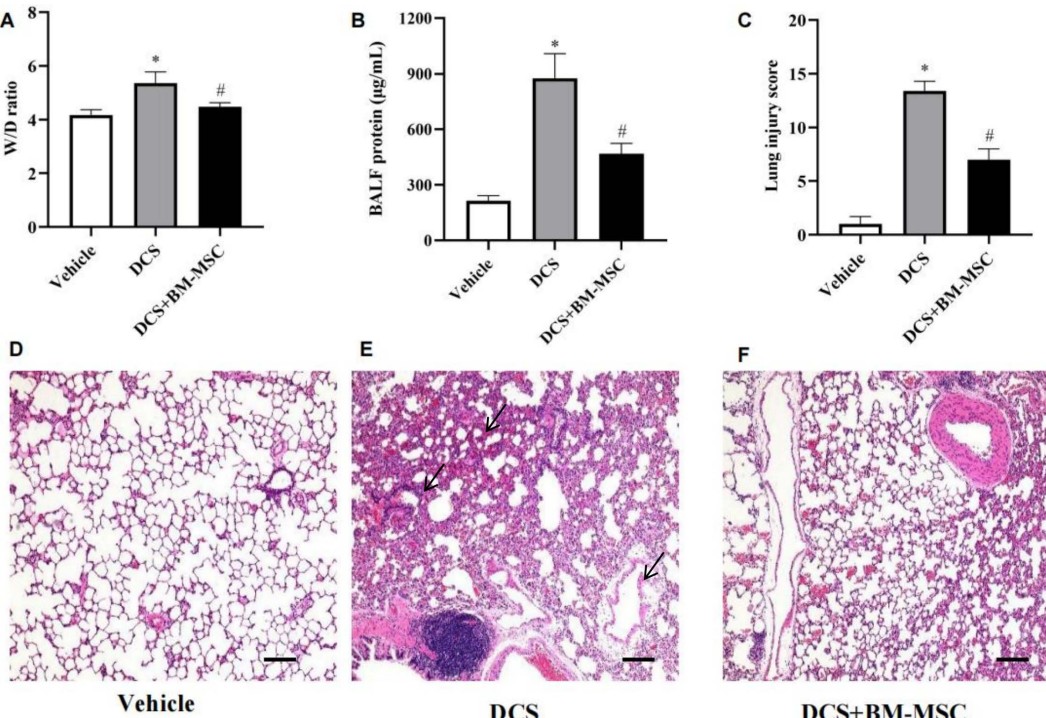

**Fig 5. Effect of BM-MSCs in improving DCS induced lung injury. (A)** Lung W/D ratio compare DCS rats.0.8800±0.1975 (n=5, 95% CI: −1.335 to −0.4246, p<0.05). **(B)** BALF protein compare DCS rats −407.0±64.53 (n=5, 95% CI: 1.335 to −0.4246, p<0.05) compare DCS rats. **(C)** Mean lung injury score compare DCS rats −6.400±0.6000 (n=5, 95% CI: 1.335 to −0.4246, p<0.05) **(D-F)** The present picture of lung tissue histopathology. *n=5 * p<0.05* vs. vehicle group. # *p<0.05* vs. DCS group. Scale bar=100 μm.

## Discussion

DCS is a fatal disease especially for divers, astronauts, and pilot. In practice, DCS takes place even when decompression tables is adhered [34,35]. Moreover, while hyperbaric oxygen is considered as the effective treatment of DCS, standard hyperbaric oxygen treatment may be delayed when decompression sickness occurs in distant places. As high mortality and morbidity of DCS, it has garnered significant attention.

Haldane model was first documented decompression model and from that on various DCS model algorithms were put forward [36–38]. We take previous study as reference then put forward a newly reliable DCS model as described above [5]. In addition, during the process of model research, we find that mortality and incidence are solidly relate to pressure and decompression rate. As far as it is known, this is the first study investigating the relationship between mortality and pressure rate. Besides, we found that injecting BM-MSCs 1h before decompression could significantly reduce death rate, while injecting 7d and 3d before did not influence it. It is interesting, as documented MSCs could survive for weeks. This result we doubt may attribute to BM-MSCs already differentiation into other cells before decompression.

DCS affect multi-system of which respiratory system is devastating influence. Former studies have demonstrated that once DCS caused lung injury usually means poor prognosis, and therefore DCS induced lung injury need further study. Our results indicate that BM-MSCs preconditioning protect against DCS induced lung injury by reducing inflammatory and oxidative stress, decreasing pyroptosis and ferroptosis, promoting survival in DCS injury. DCS patients and rodents display respiratory distress and lung injury [35]. Our present results confirmed that DCS rats had lung dysfunction evidenced by increased W/D and lung damage scores. Pretreatment with BM-MSCs significantly reversed the DCS-induced

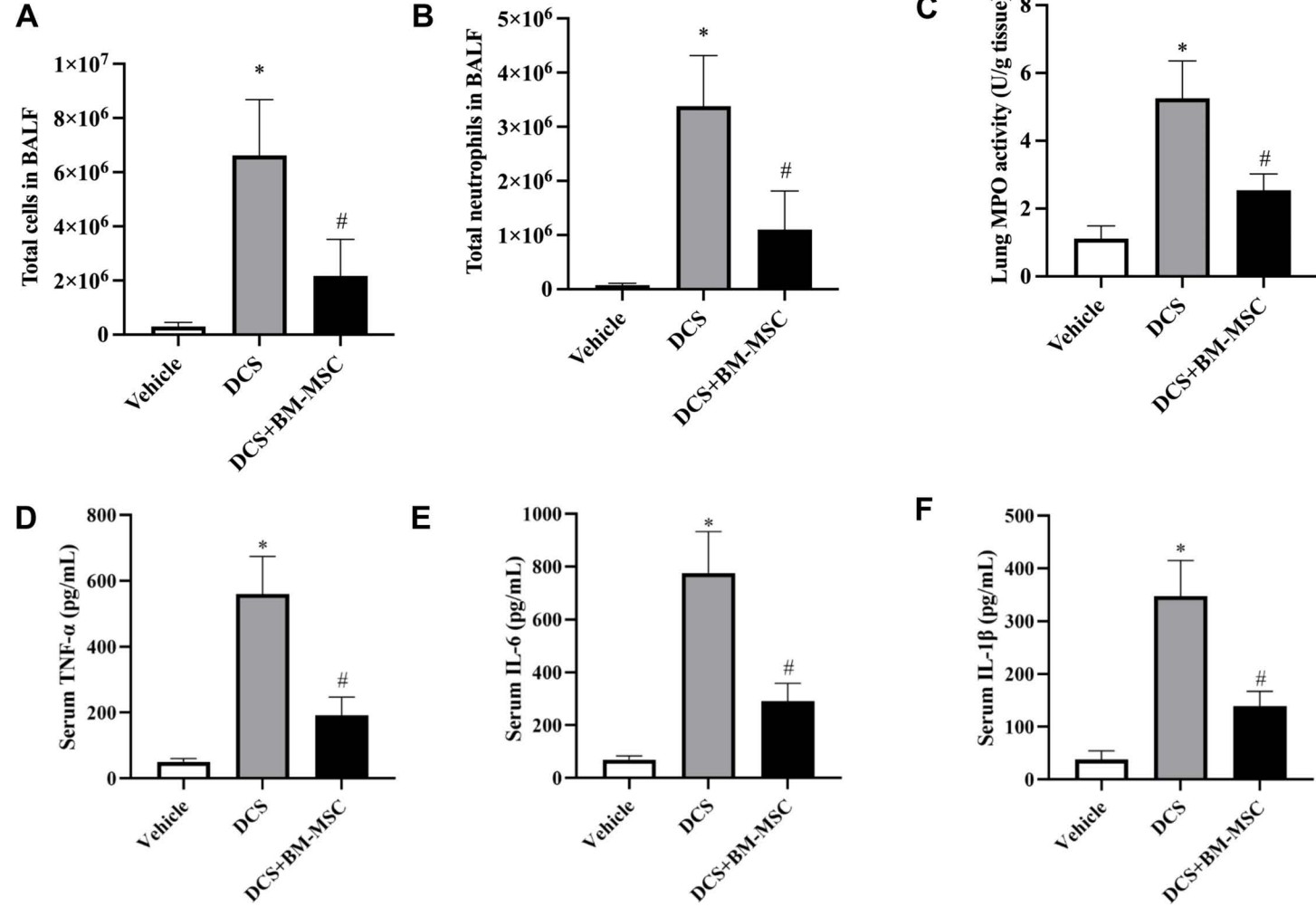

**Fig 6. Pretreatment of BM-MSCs attenuates DCS induced lung inflammation. (A)** Total cells in BALF $-4.45*10^6 \pm 1.10*10^6$ (n = 5, 95% CI: $-6.99*10^6$ to $-1.91*10^6$, p < 0.05) compare with DCS group. **(B)** Total neutrophil counts in BALF $-2.28*10^6 \pm 0.5*10^6$ (n = 5, 95% CI: $-3.49*10^6$ to $-1.1*10^6$, p < 0.05) compare with DCS group **(C)** Lung MPO activity $-2.720 \pm 0.5359$ (n = 5, 95% CI: $-3.956$ to $-1.484$, p < 0.05) compare with DCS. **(D)** Serum TNF-α level reduce $-369.4 \pm 56.67$ (n = 5, 95% CI: $-500.1$ to $-238.7$, p < 0.05) compare with DCS. **(E)** Serum IL-6 level $-484.6 \pm 76.36$ (n = 5, 95% CI: $-660.7$ to $-308.5$, p < 0.05) compare with DCS. **(F)** Serum IL-1β reduce $-484.6 \pm 76.36$ (n = 5, 95% CI: $-660.7$ to $-308.5$, p < 0.05) compare with DCS. n = 5, * p < 0.05 vs. vehicle group. # p < 0.05 vs. DCS group.

lung injury. Our present results consist with many previous researches. For example, BM-MSCs protects lung from LPS injury [16]. MSCs infusion ameliorate cigarette smoke-induced lung damage in chronic obstructive pulmonary disease [39]. BM-MSCs exosomes derived from mesenchymal stem cells possess protective effects in radiation-induced lung fibrosis [40]. MSCs is characterized by antioxidative effect, which is proved in previous studies [41,42]. BM-MSCs scavenge overproduction reactive oxygen species [43], and increase the content of anti-oxidative material, for example SOD [44], CAT [42], and GSH to alleviate oxidative injury, decrease MDA [44] and MPO [45]. Our present data further demonstrated that BM-MSCs not only promote activities of the anti-oxidant enzymes (such as SOD, CAT and GSH), but also decrease generation of ROS, MPO and MDA in DCS rat. Thus, BM-MSCs may attenuate lung injury in DCS rats via its anti-oxidative mechanisms. While excessive ROS contributes to cellular damage (e.g., DNA, lipids, and protein oxidation), low to moderate ROS levels are essential for cellular signaling, immune responses [46]. Therefore, over-suppression of

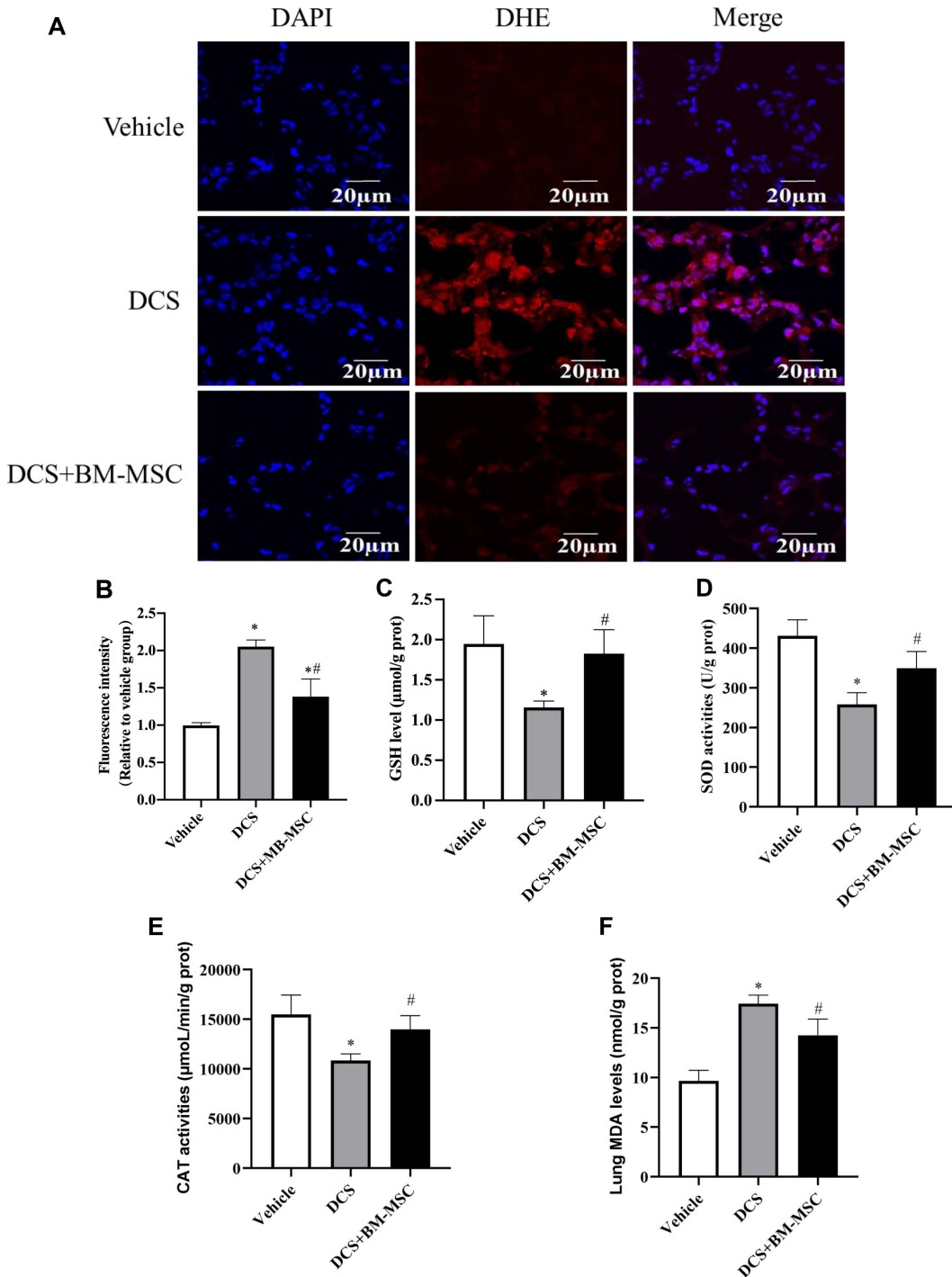

**Fig 7. Effects of BM-MSCs on ROS generation in DCS rats. (A)** Representative fluorescent images of lung tissue different groups. **(B)** Quantitative analysis of DCS-induced ROS generation of lung tissue in different. **(C)** CAT activities in different group. **(D)** GSH level in different group. **(E)** SOD activities in different group. **(F)** MDA level in different group. $n = 5$ * $p < 0.05$ vs. vehicle group. # $p < 0.05$ vs. DCS group.

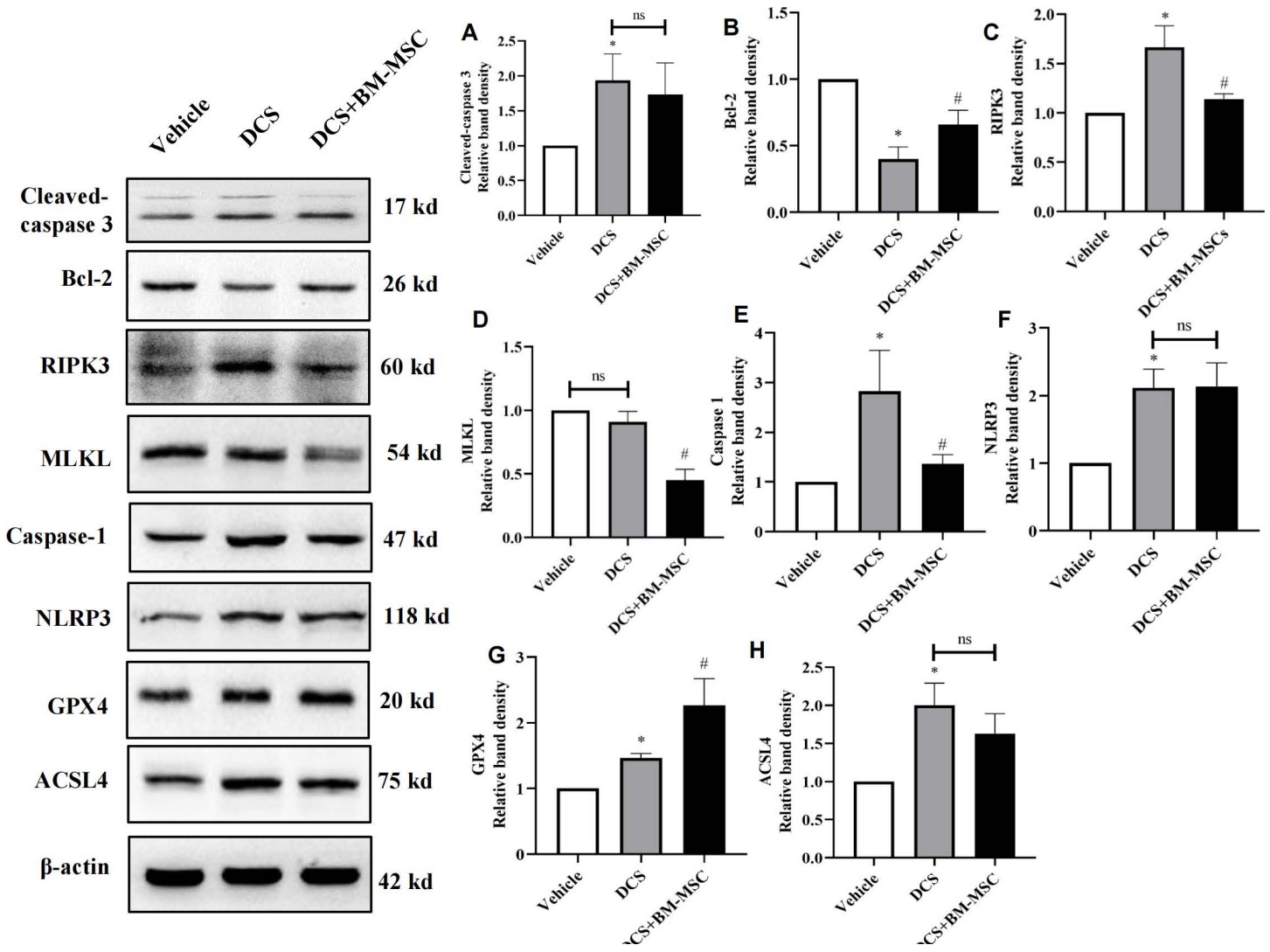

**Fig 8. Effects of BM-MSCs on different types of lung cell death in different group. (A)** Representative western-blot images of lung tissue in different groups. **(B-I)** The relative band intensities of three group are shown in histograms above. $n = 5$ * $p < 0.05$ vs. vehicle group. #$p < 0.05$ vs. DCS group.

ROS may disrupt these beneficial pathways, leading to unintended consequences such as impaired immune function or reduced stress adaptation. Even with ROS reduction, pre-existing oxidative damage (e.g., accumulated oxidized LDL in atherosclerosis or protein aggregates in aging cells) may persist. For example, antioxidant therapies in atherosclerosis trials often fail to reverse endothelial dysfunction fully, as oxidized biomolecules like ox-LDL continue to drive inflammation. ROS reduction alone may not address downstream inflammatory cascades. In kidney injury models, ROS-responsive nanoparticles (RBCM@CeO2/TAK-242) were effective only when combined with anti-inflammatory agents (e.g., TAK-242) that suppress M1 macrophage polarization and TLR4/NF-κB signaling. Similarly, atherosclerosis involves ROS-triggered endothelial dysfunction and immune cell activation, requiring multi-targeted interventions. In conclusion, BM-MSCs mitigate DCS rats injuries in both short and long term not only just by decreasing ROS, but also combinate with others mechanism.

BM-MSCs display anti-inflammatory effect by reducing IL-6, TNF-α and NF-κB, while there were increased levels of IL-10 in osteoarthritis patients [47]. In one study, BM-MSCs attenuate sepsis by prostaglandin E2–dependent reprogramming of host macrophages [48]. In another study, administration of MSC protected the airways from allergen-induced pathology, reducing airway inflammation [49]. Our study further demonstrated that DCS could cause inflammation as evidenced by significantly elevate of IL- 1β, IL-6, and TNF-α in DCS rat, while MB-MSCs attenuated lung over-production of pro-inflammatory cytokines such as TNF-α, IL-1β and IL-6 in decompression rats. These observations tend to support the hypothesis that BM-MSCs may ameliorate DCS induced lung injury via inhibiting inflammation response.

Each day billions of cells die and clear up by phagocytes. This process operates smoothly under normal conditions and therefore guarantee body function. However, this system can collapse when massive of cells suddenly die in some special conditions, such as during inflammation, and tissue damage. As we mentioned above, DCS arouse inflammation and cause tissue damages, and therefore it is likely induced cell death. Former study mainly focused on inflammation or oxidative stress of DCS, but few studies have ever found out which type of cell death involved in process of DCS and whether BM-MSCs could ameliorate DCS induced lung injury. Our current results reinforce that apoptosis is process of DCS as evidenced by increase of Cleaved-Caspase 3 and decrease of Bcl-2, which is consistent with former study [50], in present study we found pretreatment BM-MSCs have a trend of reduce apoptosis although it does not achieve statistics significance. Former study suggested that MSCs could improve smoke induced lung injury by reducing apoptosis and it seems that pluripotent stem cell-derived MSCs is superior over BM-MSCs as iPSC-MSCs possess a greater expansion capacity [51] This result is in contrast to our results, and several factors may contribute to discrepancy: [1] BM-MSCs number different. Previous study pretreatment subject $3 \times 10^6$ cells of BM-MSCs, but our study precondition the rats $2 \times 10^6$ cells of BM-MSCs. [2] Model different. Apoptosis may play important role in CS-induced lung injury model. While apoptosis may also involve in DCS induced lung injury, it may not play a key factor in lung injury. Therefore, whether BM-MSC could reduce apoptosis is still need further study.

Necroptosis, different from apoptosis, is another type of programed cell death involves cell swelling, membrane rupture, and release of cytoplasmic contents [52]. Necroptosis induced by reactive oxygen species and has been demonstrated in several animal models of diseases, such as LPS-induced lung injury, respiratory syncytial virus induced lung injury and Ischemia-Reperfusion Injury [53–55].However it is still poorly understood whether necroptosis is involved in DCS induced lung injury. In current study, we found though RIPK3 is increased in the lung of DCS rat, it could not eventually lead to rising of MLKL. Those results suggest that necroptosis may not involve in DCS, at least not through MLKL, as evidenced by contend of MLKL is not significant difference between Vehicle group and DCS group. The discrepancies in the expression levels of RIPK3 and MLKL relative to lung injury severity can be explained by their distinct roles in necroptosis signaling and context-dependent regulatory mechanisms. RIPK3 acts both as a kinase (phosphorylating MLKL to execute necroptosis) and a scaffold protein. For example, in pseudomonas aeruginosa pneumonia, RIPK3 promotes lung inflammation through its scaffold domain and RHIM motif, even when its kinase activity is inactive. This explains why RIPK3 levels may not always correlate with MLKL activation or cell death.[56]. It is worth mentioning former study also found RIPK3 increased in DCS induced lung injury, but it did not further examine the contend of MLKL [5]. As DCS related study is rare, it demands further research to verify.

Pyroptosis is recently identified programmed cell death and Caspase-1 is a crucial protease mediated process of cell death [57]. Pyroptosis is different from other cell death, has unique character, which manifests as cells continuing to expand until the membrane ruptures, causing the release of contents and activating a strong inflammatory response. Inflammatory cytokine release cause tissue damage, and tissue damage in turn aggravate inflammation. Caspase-1 dependent Pyroptosis is reported in many diseases [58]. One study has demonstrated that MSCs alleviated post-resuscitation cardiac and cerebral injuries in swine by inhibition of cell pyroptosis [59] One vitro study demonstrated that MSCs reduce Pyroptosis after ischemic stroke by targeting absent in melanoma 2. Inflammation plays important role

in DCS, and pyroptosis is also characterized by inflammation. Currently, we found that DCS may cause lung cell pyroptosis by activation NLRP3/Caspase-1 signaling pathway. Interestingly, we found BM-MSCs may inhibit pyroptosis as evidence by decrease of caspase1, but it seems not through direct decrease NLRP3.

Ferroptosis, a new type of programmed cell death, has been discovered in numerous human diseases [60]. Iron overload and reactive oxygen species play crucial roles in Ferroptosis. It was reported that BMSC-derived exosomes protect liver cell Ferroptosis by reduction of lipid peroxidation [61]. Another study reported that BM-MSCs ameliorate Ferroptosis by reduce the excessive mitochondrial fission and mitophagy, restored the mitochondrial quality control [62]. GPX4 moonlights as structural protein and antioxidant that powerfully inhibits lipid oxidation. It is regarded as a main regulate factor of ferroptosis, which participate in the lipid metabolism and influences the cell death [63]. ACSL4 is an enzyme that esterifies CoA into specific polyunsaturated fatty acid, it is another key regulator contributes to the execution of ferroptosis by triggering phospholipid peroxidation. In current study, the level of the GPX4 and ACSL4 significantly increased in DCS rats, and BM-MSCs provided significant protection against the injuries. It is noteworthy that level of ACSL4 is not significant between DCS group and DCS+BM-MSCs, which hint BM-MSCs protect ferroptosis by augment content of GPX4 instead of diminish ACSL4 production. The study's exploration of BM-MSC-mediated modulation of pyroptosis and ferroptosis in decompression sickness (DCS)-induced lung injury represents a significant advancement in the field of diving medicine and regenerative therapy. Here's how this approach enhances scientific understanding and clinical translation:1) comprehensive analysis of cross-talk between cell death pathways in DCS. While previous studies on DCS focused primarily on oxidative stress or inflammation, this work systematically links mitochondrial ROS overproduction to the activation of multiple cell death mechanisms. For example: pyroptosis (NLRP3/Caspase-1) was identified as a dominant pathway in severe lung injury, correlating with inflammation. Ferroptosis (GPX4/ACSL4) emerged as a novel contributor to lipid peroxidation, exacerbating tissue damage during rapid decompression. This dual-pathway inhibition by BM-MSCs demonstrates their pleiotropic protective effects, surpassing single-target therapies. 2) Mechanistic synergy between apoptosis and necroptosis. The study reveals that BM-MSCs not only suppress pyroptosis but also disrupt Ferroptosis. This dual action prevents both programmed and inflammatory cell death, addressing a critical gap in DCS research where prior therapies targeted only isolated pathways. Notably, the crosstalk between these pathways amplifies lung injury, making combined inhibition essential for effective treatment. 3) Clinical relevance of multi-pathway targeting. DCS-induced lung injury involves heterogeneous cellular responses, necessitating therapies with broad-spectrum efficacy. BM-MSCs' ability to: Attenuate pyrotosis -driven inflammation. Rescue Ferroptosis-mediated lipid peroxidation validates their potential as a "pan-cell death" therapeutic agent. This is particularly impactful for SCUBA divers, which rapid decompression triggers multifaceted pathology.

Our findings demonstrate the efficacy of BM-MSCs as a preventive potential. However, we have to acknowledge certain limitations of present work. First, as the initiation of DCS is obscure and hard to observe clinically. When signs and symptoms of DCS can be detected, it may be difficult to stop the development of DCS. Whether BM-MSCs has therapeutic benefit when DCS is already developed is not clear. A further limitation is current study demonstrates BM-MSCs improve DCS induced acute lung injury, whether DCS could induced chronic lung injury and BM-MSCs has same effects on that need further research to elucidate.

## Conclusion

Our results suggested that BM-MSCs preventive potential in rats subjected to DCS induced acute lung injury. Furthermore, histopathological evaluation suggested that BM-MSCs markedly improving lung injury compared with without given BM-MSCs. In current study we found that BM-MSCs appears to ameliorate DCS induced lung injury via reducing Pyroptosis, attenuating Ferroptosis, lessening oxidative stress and alleviating inflammation. The mechanisms of how BM-MSCs alleviate decompression sickness induced acute lung injury are presented in Fig 9. Prophylactic use limits clinical translation; therapeutic trials are needed post-DCS onset.

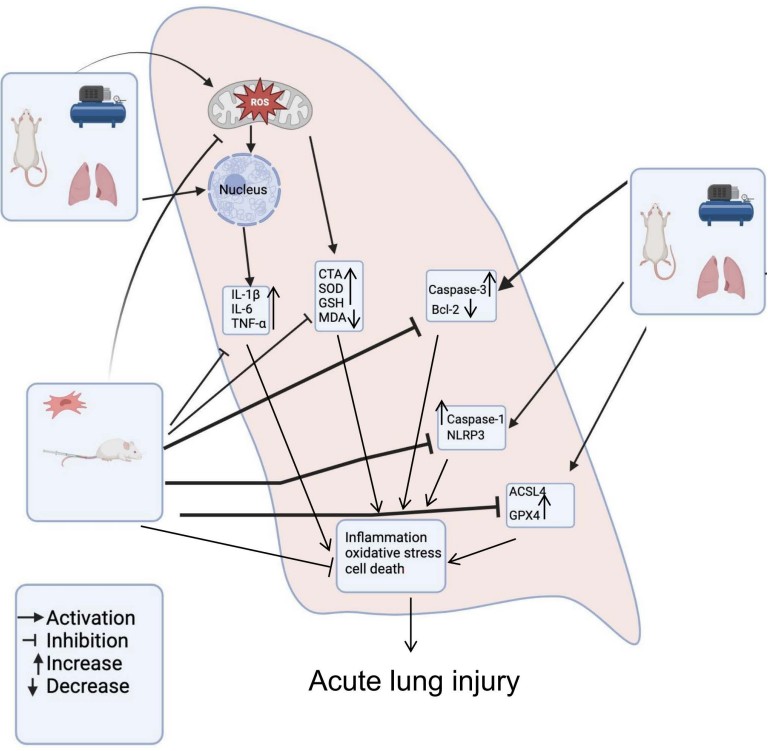

**Fig 9. A schematic representation of how BM-MSCs reduce inflammation, oxidative stress, and cell death to alleviate decompression induced acute lung injury.**

## Supporting information

**S1 File. Raw Data.**

(XLSX)

**S2 File. Original uncropped blot.**

(TIF)

## Author contributions

**Conceptualization:** Daqian Gu.

**Data curation:** Chen Lu, Jie Chen, Yuwei Weng.

**Formal analysis:** Chen Lu, Hao Chen.

**Methodology:** Liang Chen.

**Writing – original draft:** Chen Lu, Daqian Gu.

**Writing – review & editing:** Daqian Gu, Xianliang Lin.

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
