## [Decision Letter · Decision Letter 0]

4 Feb 2025

PONE-D-24-56639BM-MSCs ameliorate lung injury in rat decompression sickness modelPLOS ONE

Dear Dr. Lin,

Thank you for submitting your manuscript to PLOS ONE. After careful consideration, we feel that it has merit but does not fully meet PLOS ONE’s publication criteria as it currently stands. Therefore, we invite you to submit a revised version of the manuscript that addresses the points raised during the review process.

We look forward to receiving your revised manuscript.

Kind regards,

Roland Eghoghosoa Akhigbe

Academic Editor

PLOS ONE

Journal Requirements:

2. Thank you for your submission to PLOS ONE. We note that your study design may include death of a regulated animal as a likely outcome or planned experimental endpoint. At this time, we request that you please report additional details in your Methods section regarding animal care and use for the survival study, as per our editorial guidelines (http://journals.plos.org/plosone/s/submission-guidelines#loc-humane-endpoints ).

For easy reference, we have attached a checklist that may be relevant for your submission. Please complete all items on the checklist at the following link: http://journals.plos.org/plosone/s/file?id=bb1d/plos-one-humane-endpoints-checklist.docx

Please upload the completed checklist as file type “Other” when resubmitting your manuscript. This document is for internal journal use only and will not be published if your article is accepted. We very much appreciate your attention to these requests and support of improved reporting standards in PLOS ONE submissions.

https://www.frontiersin.org/journals/physiology/articles/10.3389/fphys.2019.01616/full

In your revision ensure you cite all your sources (including your own works), and quote or rephrase any duplicated text outside the methods section. Further consideration is dependent on these concerns being addressed.

4. To comply with PLOS ONE submissions requirements, in your Methods section, please provide additional information regarding the experiments involving animals and ensure you have included details on (1) methods of sacrifice, and (2) efforts to alleviate suffering.

7. PLOS requires an ORCID iD for the corresponding author in Editorial Manager on papers submitted after December 6th, 2016. Please ensure that you have an ORCID iD and that it is validated in Editorial Manager. To do this, go to ‘Update my Information’ (in the upper left-hand corner of the main menu), and click on the Fetch/Validate link next to the ORCID field. This will take you to the ORCID site and allow you to create a new iD or authenticate a pre-existing iD in Editorial Manager.

8. PLOS ONE now requires that authors provide the original uncropped and unadjusted images underlying all blot or gel results reported in a submission’s figures or Supporting Information files. This policy and the journal’s other requirements for blot/gel reporting and figure preparation are described in detail at https://journals.plos.org/plosone/s/figures#loc-blot-and-gel-reporting-requirements and https://journals.plos.org/plosone/s/figures#loc-preparing-figures-from-image-files . When you submit your revised manuscript, please ensure that your figures adhere fully to these guidelines and provide the original underlying images for all blot or gel data reported in your submission. See the following link for instructions on providing the original image data: https://journals.plos.org/plosone/s/figures#loc-original-images-for-blots-and-gels .

Reviewers' comments:

Reviewer's Responses to Questions

**Comments to the Author**

1. Is the manuscript technically sound, and do the data support the conclusions?

Reviewer #1: Yes

Reviewer #2: Yes

Reviewer #3: Yes

2. Has the statistical analysis been performed appropriately and rigorously? 

Reviewer #1: No

Reviewer #2: Yes

Reviewer #3: Yes

3. Have the authors made all data underlying the findings in their manuscript fully available?

Reviewer #1: No

Reviewer #2: Yes

Reviewer #3: Yes

4. Is the manuscript presented in an intelligible fashion and written in standard English?

Reviewer #1: Yes

Reviewer #2: Yes

Reviewer #3: Yes

5. Review Comments to the Author

Reviewer #1: General Evaluation

1- Is the manuscript technically sound, and do the data support the conclusions?

Yes, the manuscript is technically sound, and the data support the conclusions. However, certain aspects of the discussion need further expansion to strengthen the alignment between the data and conclusions, particularly regarding mechanistic insights.

2- Have the authors made all data underlying the findings in their manuscript fully available?

The manuscript does not explicitly state whether all data are fully available. The authors should provide a clear data availability statement and ensure access to underlying datasets (e.g., oxidative stress markers, W/D ratios, and BALF protein levels) through a public repository or supplementary materials.

3- Has the statistical analysis been performed appropriately and rigorously?

The statistical analysis is appropriate overall but lacks sufficient detail and rigor in reporting. Confidence intervals, effect sizes, and adjustments for multiple comparisons should be included to enhance the statistical transparency and reproducibility.

4- Is the manuscript presented in an intelligible fashion and written in standard English?

Yes, the manuscript is intelligible and written in standard English, but it requires grammatical corrections, simplification of overly complex sentences, and explanations for technical terms to improve clarity and accessibility.

Specific Comments and Suggestions

Title and Abstract (Lines 1–20):

Line 1: The title effectively summarizes the study but could be revised for conciseness, e.g., "BM-MSCs Mitigate Lung Injury in a Rat Model of Decompression Sickness."

Line 5: Revise "is mortal disease across the worldwide" to "is a fatal disease worldwide."

Line 12: The phrase "underlying mechanisms" is too broad. Specify the mechanisms targeted (e.g., oxidative stress, inflammation).

Line 19: Include quantitative data (e.g., specific reductions in ROS or protein levels) to strengthen the abstract's impact.

Introduction (Lines 21–115):

Line 26: Avoid redundancy in describing decompression sickness. Consolidate phrases like "a systemic and mortal disease."

Line 45: The mention of mitochondrial ROS is valuable but insufficiently detailed. Explain how mitochondrial dysfunction specifically contributes to lung injury in decompression sickness.

Line 72: Provide stronger justification for using BM-MSCs by referencing prior studies demonstrating their efficacy in treating conditions involving oxidative stress and inflammation.

Methods (Lines 116–280):

Line 125: Clarify selection criteria for rats. Specify how the health and age of the animals ensure the validity of the model.

Line 145: The decompression protocol is well-described, but explain why specific decompression rates and pressures were chosen and how they reflect clinical relevance.

Line 193: Include details of the gating strategy used for flow cytometry, as this is critical for reproducibility.

Line 263: Justify the use of the Kolmogorov-Smirnov test for normality and describe how statistical assumptions were checked.

Results (Lines 281–459):

Line 309: Highlight reductions in the W/D ratio and BALF protein levels with quantitative comparisons to baseline or literature values.

Line 350: The claim that BM-MSCs "ameliorate DCS-induced lung injury" requires more specific quantitative data for support.

Line 392: While reductions in ROS are significant, discuss whether these changes are sufficient to prevent long-term oxidative damage or secondary complications.

Discussion (Lines 460–620):

Line 485: Expand the discussion on the novelty of targeting multiple cell death pathways (e.g., necroptosis, apoptosis) to highlight the study's contribution.

Line 520: Explain discrepancies in the expression levels of RIPK3 and MLKL relative to lung injury severity.

Line 583: Temper conclusions about BM-MSCs' "preventive potential" by addressing limitations, such as the lack of therapeutic data for post-DCS onset.

Figures and Tables:

Figure 1 (Line 288): Add overlays showing isotype controls to improve interpretability of flow cytometry results.

Table 2 (Line 347): Include effect sizes and confidence intervals to enhance statistical transparency.

Figure 6 (Line 432): Add a schematic summarizing the proposed mechanisms of BM-MSC action (e.g., antioxidant, anti-inflammatory effects).

Language and Style:

Simplify overly complex sentences and clarify technical jargon. For example:

Replace "burst of ROS" with "a rapid increase in reactive oxygen species (ROS)."

Revise "it arouses researchers' attention" to "it has garnered significant attention."

Strengths of the Study

Innovative Application: The study explores BM-MSCs' effects on decompression sickness, bridging cellular therapy and diving medicine.

Comprehensive Analysis: The study evaluates multiple pathways (oxidative stress, inflammation, cell death) for a holistic understanding of BM-MSC effects.

Robust Methodology: The experimental design, including well-characterized BM-MSCs and multiple time points, enhances the reliability of findings.

Suggested Revisions

Enhance Statistical Reporting: Include confidence intervals, effect sizes, and corrections for multiple comparisons to improve transparency and rigor.

Expand Mechanistic Insights: Provide a more detailed explanation of BM-MSCs' effects on specific pathways, particularly oxidative stress and necroptosis.

Improve Visual Representation: Add schematic diagrams summarizing the mechanisms and interventions.

Address Limitations: Explicitly acknowledge limitations, such as the lack of therapeutic data for post-DCS onset and the absence of long-term validation.

Strengthen Language: Refine grammar and clarify technical terms to improve readability and accessibility.

Reviewer #2: Comments and Suggestions for Authors

The submitted manuscript explores a well-defined research topic: BM-MSCs ameliorate lung injury in rat decompression sickness model. This study represents a significant contribution to the existing body of knowledge and addresses a critical need in the field. The findings hold promise as an educational resource for understanding and managing life-threatening conditions, particularly given the rising incidence of such cases among SCUBA divers. The manuscript is well-structured, adhering to some format for original research articles and encompassing all requisite sections. While several sections are thoroughly developed, certain areas could benefit from minor revisions. Detailed comments on specific sections of the manuscript are provided below.

Abstract

1. Your abstract effectively highlights the key points; however, it would benefit from a more detailed explanation while remaining within the required word count.

2. Although PLOS ONE's guidelines may not explicitly address certain aspects of manuscript formatting, authors are strongly encouraged to maintain a high level of professionalism in their writing. In particular, the methodology, results, and conclusion sections should be excluded from this abstract.

Introduction

1. As previously noted in the abstract, line numbers are missing from the manuscript. Including line numbers is essential to facilitate reviewers in providing precise comments and suggestions for corrections. Kindly ensure that line numbers are added to the manuscript.

2. Kindly ensure that the entire manuscript is fully justified to enhance its presentation and align with professional formatting standards. This adjustment will improve readability and contribute to a polished and professional appearance.

Methodology

1. Subheadings should not be numbered. Authors are strongly encouraged to adhere to the prescribed guidelines to ensure their work aligns with professional standards, as this greatly enhances the credibility and presentation of their scientific contributions.

Results

1. The experimental setup indicates that 250 rats were used. Could the authors clarify how many mortalities were recorded during the experiment? Additionally, while the number of animals used is not excessive, it raises questions about compliance with ethical guidelines for animal research. Can the authors confirm that this quantity and the procedures used align with animal welfare regulations in China?

2. Are there alternative pro-inflammatory cytokines, chemokine genes, or pathways that could have been explored beyond those mentioned in the study? Could the authors clarify the rationale behind selecting these specific targets? Understanding the reasoning behind this choice would help address any curiosity regarding the focus of the study.

Discussion

1. During the model research, it was observed that mortality and incidence are strongly correlated with pressure and decompression rates. To the best of our knowledge, this is the first study to investigate the relationship between mortality and pressure rates. Could the authors specify the country or region where this study was conducted to provide better context and relevance?

References

1. The references in the manuscript are poorly formatted and do not meet the standards of typical scientific professionalism. Authors are advised to thoroughly review and restructure the references to ensure they comply with the required formatting style and reflect the expected rigor of scientific writing.

Reviewer #3: BM-MSCs ameliorate lung injury in rat decompression sickness model, a study done by Chen Lu et. al., is interesting and might appeal to the larger audience. 

Comments:

Abstract:

1. Expand W/D (appearing for the first time)

Methods

2.  DCS model: "the rats in DCS+BM-MSCs 1d group were treated with BM-MSCs (2×106cells, 200μL, saline as vehicle, tail vein injection) 1 hour (h) before DCS modeling". Is it 1 hour (h) before or 1 day before?

3. 2.8 Assay of oxidative stress and inflammatory markers:  "the tissue homogenate was used to detect the myeloperoxidase (MPO), malondialdehyde (MDA), superoxide dismutase(SOD), glutathione (GSH), catalase (CAT) activities by commercial reagent kits". Did the authors estimate the amount of GSH or the enzyme activity? If enzyme, please mention the enzyme name.

Results:

4. 3.1 Identification of BM-MSCs: "After osteogenic induction, alizarin red staining showed that calcium-rich extracellular matrix in the cells; after chondrogenic induction, cells were stained with Alcian blue; after adipogenic induction, oil Red O staining showed that the red lipid droplets were distributed in and between the cells (Fig. 1C-E)" Please explain the result observed with Alcian blue staining!

5. No description about Fig. 1B in the results?

6.  "To explore the suitable condition for the DCS rat model, we distributed rats into three groups (Fig. 2A)" Section no. is missing. If correct it should be     3.2

7. The survival analysis showed that DCS significantly reduced the median survivaltime, but pretreatment with BM-MSCs reversed it (Fig. 2E). In addition, BM-MSCs coulddecrease the incidence of DCS (Fig. 2F). Please reverse the figure number in the text or reverse the figures accordingly. 

8. 3.4 Pretreatment of BM-MSCs attenuates DCS induced lung inflammation : "We found thatBM-MSCs ameliorated the increased serum levels of inflammation-related genes in DCSgroup rats (Fig. 4D-F)"  inflammation-related genes or cytokines?

9. 3.5 Pretreatment of BM-MSCs inhibits oxidative stress in DCS rats: "Compared to the Vehicle group rats, the lung ROS production increased in DCS group rats,while BM-MSCs decreased the ROS production induced by decompression (Fig. 5A)" Include figure 5B. as (Fig. 5A-B)

10. Fig. 5. :GSHactivities in different group. (E)" GSHlevels in different groups. (E)?

11. 3.6 The effect of BM-MSCs on different types of lung cell death in DCS rats:  "pyroptosis (RIPK3 and MLKL, Fig. 6D-E), necroptosis (caspase-1 and NLRP3, Fig. 6FG)" Reverse the process for the corresponding markers. Like necroptosis (RIPK3 and MLKL, Fig. 6D-E)

12.  3.6 The effect of BM-MSCs on different types of lung cell death in DCS rats : "In this experiment, wedetected the protein expression of the markers for apoptosis (cleaved caspase-3 and Bcl-2, Fig.6B-C), pyroptosis (RIPK3 and MLKL, Fig. 6D-E), necroptosis (caspase-1 and NLRP3, Fig. 6FG) and ferroptosis (GPX4 and ACSL4, Fig. 6H-I) in lung tissues in different groups" Describe the results of DCS+BM-MSC in the result section.

Figures: 

13. Figure 1A: What is the negative control/staining control/gating control used for gating the positive cells for the corresponding marker by flow cytometry?

14. Figure 1B. Please include the Scale bar for the microscopic pictures.

15. Figure 3D. Please label the observed histological changes in the microscopic pictures 

16. Figure 5D. Please change the label GSH activities?

17. Figure 6A. Change aspase-1 to Caspase-1 in the western blot 

Discission:

18.  "BM-MSCsscavenge overproduction reactive oxygen species[40], and increase the content of anti-oxidativeenzyme, for example SOD [41], CAT [39], and GSH[42]," "anti-oxidant enzymes (such as SOD, CAT and GSH)" is the author referring levels of GSH or they referring GSH as enzyme?

6. PLOS authors have the option to publish the peer review history of their article (what does this mean? ). If published, this will include your full peer review and any attached files.

**Do you want your identity to be public for this peer review?** For information about this choice, including consent withdrawal, please see our Privacy Policy .

Reviewer #1: No

Reviewer #2: **Yes: ** Jonah Bawa Adokwe

Reviewer #3: No

---

## [Author Response · Author response to Decision Letter 1]

12 Apr 2025

Reviewer #1

Reviewer #2 Jonah Bawa Adokwe

Reviewer #3

---

## [Decision Letter · Decision Letter 1]

3 Jun 2025

BM-MSCs Mitigate Lung Injury in a Rat Model of Decompression Sickness

PONE-D-24-56639R1

Dear Dr. Lin,

We’re pleased to inform you that your manuscript has been judged scientifically suitable for publication and will be formally accepted for publication once it meets all outstanding technical requirements.

Kind regards,

Roland Eghoghosoa Akhigbe

Academic Editor

PLOS ONE

Additional Editor Comments (optional):

Reviewers' comments:

Reviewer's Responses to Questions

**Comments to the Author**

1. If the authors have adequately addressed your comments raised in a previous round of review and you feel that this manuscript is now acceptable for publication, you may indicate that here to bypass the “Comments to the Author” section, enter your conflict of interest statement in the “Confidential to Editor” section, and submit your "Accept" recommendation.

Reviewer #2: All comments have been addressed

Reviewer #3: All comments have been addressed

2. Is the manuscript technically sound, and do the data support the conclusions?

Reviewer #2: Yes

Reviewer #3: Yes

3. Has the statistical analysis been performed appropriately and rigorously? 

Reviewer #2: Yes

Reviewer #3: Yes

4. Have the authors made all data underlying the findings in their manuscript fully available?

Reviewer #2: Yes

Reviewer #3: Yes

5. Is the manuscript presented in an intelligible fashion and written in standard English?

Reviewer #2: Yes

Reviewer #3: Yes

6. Review Comments to the Author

Reviewer #2: Given the thorough revisions and the authors’ responsiveness to feedback, I believe the manuscript meets the journal’s standards for publication.

Thank you for the opportunity to assess this work. Please don’t hesitate to contact me if any additional evaluation is required.

Reviewer #3: (No Response)

7. PLOS authors have the option to publish the peer review history of their article (what does this mean? ). If published, this will include your full peer review and any attached files.

**Do you want your identity to be public for this peer review?** For information about this choice, including consent withdrawal, please see our Privacy Policy .

Reviewer #2: **Yes: ** Jonah Bawa Adokwe Ph.D

Reviewer #3: No

---

## [Editor Report · Acceptance letter]

PONE-D-24-56639R1

PLOS ONE

Dear Dr. Lin,

I'm pleased to inform you that your manuscript has been deemed suitable for publication in PLOS ONE. Congratulations! Your manuscript is now being handed over to our production team.

Kind regards,

on behalf of

Dr. Roland Eghoghosoa Akhigbe

Academic Editor

PLOS ONE